# The Emerging Role of Nerves and Glia in Colorectal Cancer

**DOI:** 10.3390/cancers13010152

**Published:** 2021-01-05

**Authors:** Simone L. Schonkeren, Meike S. Thijssen, Nathalie Vaes, Werend Boesmans, Veerle Melotte

**Affiliations:** 1Department of Pathology, GROW-School for Oncology and Developmental Biology, Maastricht University Medical Center, 6229 HX Maastricht, The Netherlands; simone.schonkeren@maastrichtuniversity.nl (S.L.S.); m.thijssen@maastrichtuniversity.nl (M.S.T.); n.vaes@maastrichtuniversity.nl (N.V.); werend.boesmans@uhasselt.be (W.B.); 2Biomedical Research Institute (BIOMED), Hasselt University, 3950 Diepenbeek, Belgium; 3Department of Clinical Genetics, Erasmus MC University Medical Center, 3015 GD Rotterdam, The Netherlands

**Keywords:** tumor microenvironment, colorectal cancer, neurons, glia, nerves, innervation, enteric nervous system, peripheral nervous system

## Abstract

**Simple Summary:**

The influence of nerves on different types of cancers, including colorectal cancer, is increasingly recognized. The intestines are highly innervated, both from outside the intestines (extrinsic innervation) and by a nervous system of their own; the enteric nervous system (intrinsic innervation). Nerves and cancer cells have been described to communicate with each other, although the exact mechanism in colorectal cancer is not yet explored. Nerves can enhance cancer progression by secreting signaling molecules, and cancer cells are capable of stimulating nerve growth. This review summarizes the innervation of the intestines and current knowledge on the role of the nervous system in colorectal cancer. Additionally, the therapeutic potential of these new insights is discussed.

**Abstract:**

The role of the nervous system as a contributor in the tumor microenvironment has been recognized in different cancer types, including colorectal cancer (CRC). The gastrointestinal tract is a highly innervated organ system, which is not only innervated by the autonomic nervous system, but also contains an extensive nervous system of its own; the enteric nervous system (ENS). The ENS is important for gut function and homeostasis by regulating processes such as fluid absorption, blood flow, and gut motility. Dysfunction of the ENS has been linked with multiple gastrointestinal diseases, such as Hirschsprung disease and inflammatory bowel disease, and even with neurodegenerative disorders. How the extrinsic and intrinsic innervation of the gut contributes to CRC is not fully understood, although a mutual relationship between cancer cells and nerves has been described. Nerves enhance cancer progression through the secretion of neurotransmitters and neuropeptides, and cancer cells are capable of stimulating nerve growth. This review summarizes and discusses the nervous system innervation of the gastrointestinal tract and how it can influence carcinogenesis, and vice versa. Lastly, the therapeutic potential of these novel insights is discussed.

## 1. Introduction

Cancer is one of the most complex diseases due to a phenomenon called tumor heterogeneity, the molecular diversity between and within tumors [1]. This complicates the search for and applicability of treatments. A major contribution to tumor heterogeneity comes from the cells and components surrounding the tumor cells, the tumor microenvironment (TME). Classically, the TME is considered to comprise fibroblasts, myofibroblasts, immune cells, adipose cells, neuroendocrine cells, vasculature, the lymphatic system, and extracellular matrix components [2]. Only in the last decade, the role of another component of the TME, namely the nervous system, has gained attention in different tumor types including gastric [3], prostate [4,5], breast [6], pancreatic [7] and colorectal cancer (CRC) [8]; which has been reviewed by Zahalka and Frenette [9]. 

Although the central nervous system (CNS) is a relatively secluded part of the nervous system, protected by the blood–brain barrier, the peripheral nervous system (PNS) emerges as ganglia that reside in relative proximity to the organs they innervate. Neuronal cell bodies of the PNS, and especially neural endings, are situated within or adjacent to the TME, which can be influenced by the tumor and vice versa [4,10,11]. 

In CRC, the mechanisms by which neurons influence carcinogenesis are largely unknown. This is striking, because the gastrointestinal (GI) tract is a highly innervated organ system with major connections to the CNS, but also contains a nervous system of its own: the enteric nervous system (ENS). The ENS is composed of neurons and glia structured in ganglia in the gut wall, which influence gut homeostasis, by regulating digestive functions such as secretion and absorption, local blood flow and gut motility [12]. The importance of enteric neurons in CRC was reviewed by Rademakers et al. in 2017 [13], focusing on the link between the ENS, inflammation, and CRC, as well as perineural invasion, and neurotransmitters involved in CRC. Limited new data have become available regarding the role of nerves and neurons in CRC. Furthermore, enteric glial cells (EGCs) are becoming more and more appreciated for their role in the above-mentioned functions of the ENS and their influence on enteric neuron activity and function [14]. A role for EGCs in CRC has also been proposed. 

In this review, we provide an overview of the extrinsic and intrinsic nervous system innervation in the GI tract, summarize current literature about the role of neuronal innervation and EGCs in CRC, and highlight the latest research and therapeutic potential involving the nervous system and CRC. 

## 2. Innervation of the Gastrointestinal Tract

Mammalian gut function largely depends on both extrinsic and intrinsic neural pathways innervating the GI tract. The extrinsic innervation, for which the neuronal cell bodies are located outside the gut, comprises both sympathetic and parasympathetic projections, as well as visceral afferents. The ENS, which is embedded in the gut wall, supplies the intrinsic innervation [15,16] (Figure 1). 

### 2.1. Extrinsic Innervation: The Sympathetic and Parasympathetic Nervous System

The neuronal cell bodies of extrinsic neurons innervating the gut are located in the brainstem, prevertebral ganglia and peripheral afferent ganglia (Figure 1). After colonization of the gut by ENS precursors, fibers from extrinsic sensory and visceromotor neurons project to the gut. These originate from vagal, dorsal root, sympathetic, and pelvic ganglia. Sympathetic fibers typically travel along blood vessels, while parasympathetic fibers enter the gut via the vagus nerve or spinal cord. During embryogenesis, vagal innervation is directed by chemoattractants and chemorepellents secreted by the developing epithelium, mesenchyme, and enteric neurons, but how spinal afferents innervate the gut during development is still largely unknown [17].

Sympathetic nerves have their cell bodies located in the sympathetic chain next to the spinal cord (the so-called paravertebral ganglia) and in prevertebral ganglia, which are located more closely to the GI tract (Figure 1). Sympathetic activation leads to the release of noradrenaline along with a release of other neurotransmitters, and generally delays intestinal transit and secretion via the inhibition of enteric (secreto) motor neurons, direct inhibition of smooth muscle cells contraction, and secretory actions, by inducing intestinal vasoconstriction. Sympathetic activity also causes the contraction of sphincters [18]. 

In contrast to the inhibitory action of sympathetic innervation, parasympathetic nerves have inhibitory and excitatory control over gastrointestinal motility, typically by releasing acetylcholine [19]. Parasympathetic nerve cell bodies are located in the brainstem and project to the stomach, small intestine, and proximal colon through the vagus nerve (Figure 1). Vagal input in the intestines occurs through the connection of a vagal fiber to multiple enteric neurons [20]. In this way, the vagus nerve can modulate ENS activity in a general manner. Sacral preganglionic neurons in the spinal cord and in the sacral parasympathetic nucleus control parasympathetic activity in the distal colon via communication with enteric neurons either directly or indirectly [19]. Both the vagus and pelvic nerve contain efferent (motor) and afferent (sensory) nerve fibers. These afferent nerves transfer sensory information from the GI tract to the CNS. Vagal afferent nerves have cell bodies located in the nodose and jugular ganglia, and project from there to the brain. Other afferent nerves enter the spinal cord (and thus the CNS) directly and are referred to as spinal afferents [21]. They have nerve endings in the mucosa, enteric ganglia, or in the muscle layer and are important for the control of gut function and GI sensation, including pain and satiety [22].

### 2.2. Intrinsic Innervation: The Enteric Nervous System

In mammals, such as humans and mice, intrinsic neuronal innervation of the gut is initiated during embryogenesis with vagal neural crest cells invading the foregut [23]. These enteric neural crest cells migrate in a rostral-to-caudal direction along the developing intestine. In addition, a relatively small group of ENS precursors originating from the sacral neural crest contributes to the colonization of the hindgut [23]. Successful ENS development requires extensive proliferation and differentiation into balanced numbers of functional enteric neuron and glial subtypes [17,24]. 

Mature enteric neurons and part of the EGCs are clustered in the myenteric and submucosal plexus, which are described as mesh-like structures embedded in the GI wall. The myenteric plexus ranges along the entire length of the GI tract, whereas the submucosal plexus is only located in the small and large intestine [16]. The ENS contains diverse neuronal and glial subtypes with varying transcriptomic, phenotypic, and functional properties [25,26,27,28]. 

The major neuronal types are motor neurons innervating the smooth muscle layers, vasodilatator neurons, secretomotor neurons, intrinsic primary afferent neurons (IPANs; intrinsic sensory neurons with Dogiel type II morphology that innervate the gut mucosa), interneurons, and intestinofugal neurons (Figure 1), which can be subdivided in different subpopulations as described in detail elsewhere [25,29,30]. 

EGCs can also be divided into multiple subtypes, mostly based on their location and morphology [26,31,32,33,34]. Besides being present in the myenteric and submucosal plexus, they can, contrary to enteric neurons, also be found in the lamina propria and muscle layers of the gut. 

Through the interplay between enteric neurons and glia, this intrinsic nervous system can control many gastrointestinal functions autonomously. An important experiment, performed in 1755, showed that an excised intestine, without any connection to the vagus nerve or the CNS, still exhibits peristaltic contractions [35]. Therefore, the ENS is sometimes also referred to as “the second brain” or “the brain of the gut”. Although the ENS is indispensable for GI function, input from the CNS is still important. There is great connectivity between the ENS and CNS, and together they integrate signals to control contractile bowel activity and secretion [16].

## 3. Nervous System Innervation in Colorectal Cancer

### 3.1. Neurons and Colorectal Cancer

Even though there is a booming interest in the role of neurons in cancer, knowledge on the role of the neural systems innervating the gut in CRC is still limited. A passive role for neurons in CRC and other cancer types has been recognized for several years in the context of perineural invasion (PNI) [36]. In this process, tumor cells use the nerve fibers as leading strands to migrate to other sites, which is linked to rapid tumor growth and metastasis [36]. In CRC, PNI was characterized as a prognostic marker for the aggressiveness of cancer and for survival rates [37,38]. PNI is correlated with poorly differentiated and high-grade colorectal tumors. Furthermore, PNI is associated with a reduced disease-free survival and can serve as an independent prognostic marker for patient outcomes [38]. Rectal cancers had a higher incidence of PNI, which correlates with their higher aggressiveness and poorer overall survival rates compared to colon cancer. It is thought that the higher incidence of PNI in the rectum is related to its denser extrinsic innervation compared to the colon [37,38]. PNI, however, is believed to arise via extrinsic nerve bundles from the PNS [39]; a physical interaction between enteric neurons and CRC cells has been described. Enteric neurons express N-cadherin and L1CAM, which were identified as important molecules for the adhesion and migration of CRC cells [40]. However, deletion of these molecules did not completely abrogate PNI, suggesting that other (unknown) molecules must be involved as well.

Active communication between cancer and neuronal cells has been described for different cancer types [9,41], identifying several ways of neuron–cancer signaling. By experimentally altering neuronal signaling, it has been shown that nerves can influence carcinogenesis. Tumor growth and cancer progression were enhanced by increasing activation of sympathetic nerves in an animal model of breast cancer [42], while reducing parasympathetic activity in the stomach inhibited gastric tumorigenesis [3]. In a genetic mouse model of intestinal cancer (*Apc^Min/+^* model), vagotomy decreased the tumor area, indicating that extrinsic innervation from the vagus nerve could promote carcinogenesis [43]. In contrast, surgical sympathetic denervation at the level of the mesenteric artery did not inhibit tumor growth in the same study, which suggests that the mode of action of specific nerves rather than neuronal activity per se affects carcinogenesis. 

Furthermore, cancer cells can signal to neuronal cells as well. They aim to guide innervation by expressing chemoattractant/-repellant and axon guidance molecules, as observed in prostate [44] and pancreatic cancer [45]. Head and neck cancer cells release vesicles (exosomes) containing EphrinB1 that can induce neurite outgrowth, and blocking of exosome release attenuates tumor innervation in vivo [46]. In addition, Amit et al. showed that head and neck cancer cells, mutant for p53, secrete the miRNA miR-43a within vesicles, which drives the reprogramming of cancer-associated sensory nerves to become adrenergic, similar to sympathetic neurons [47]. Together, this indicates that tumors are able to induce their own innervation.

Besides this unidirectional communication, specific crosstalk has also been described between cancer and neuronal cells, whereby nerves promote cancer development and cancer cells induce tumor innervation in a feedforward loop [48,49]. In pancreatic cancer, adrenergic signaling to tumor cells induces nerve growth factor (NGF) and brain-derived neurotrophic factor (BDNF) release by tumor cells, resulting in increased nerve density in the tumor area. This in turn results in increased adrenergic signaling, leading to accumulation of noradrenaline and enhanced tumor growth [48]. In gastric cancer, cholinergic nerves and Tuft cells affect tumorigenesis by releasing acetylcholine. This stimulates the expression of NGF from epithelial cells, which in turn increases nerve and Tuft cell density, thereby further enhancing acetylcholine release and neuronal innervation [49]. Interestingly, Hayakawa et al. also described an upregulation of NGF mRNA in murine CRC tissue. Moreover, overexpression of NGF in the colon epithelium, using a Villin-Cre: R26-NGF mouse line, resulted in the development of more and larger rectal tumors upon chemical (AOM/DSS) tumor induction, indicating that a similar process could take place in CRC [49]. 

So far, no specific bi-directional communication between neurons and cancer cells has been described for CRC. However, there are multiple studies depicting a role for neurotransmitters and neuropeptides in CRC, as previously summarized [13]. Since then, new studies have confirmed the roles of some of these signaling molecules, including Substance P and neurotensin as tumor-promoting signals, [50,51,52], somatostatin as a tumor-suppressing signal [53], and galanin as a potential biomarker for CRC in sera and tissue [54,55]. Moreover, new functions for some of the described neuropeptides have been elucidated. It was already established that neuropeptide Y (NPY) is produced by enteric neurons [56] and functions in the regulation of inflammation in the gut [57], but its role in CRC was largely unknown. Currently a tumor-promoting role for NPY is suggested, based on observations that NPY-deficient mice displayed reduced inflammation, polyp size, and polyp number in an inflammation model (DSS), and that NPY stimulates proliferation and reduces apoptosis of tumor cells in vitro [58]. Studies have shown that serotonin prevents DNA damage in the colon and thereby protects against early tumorigenesis [59], and that gamma-aminobutyric acid (GABA) inhibits proliferation and increases sensitivity to chemotherapy in CRC cells [60]. Importantly, these signaling molecules can also have tumor-promoting properties depending on the receptor to which they bind and the pathways they consequentially activate or suppress [13]. 

Given that many neurotransmitters and neuropeptides have multiple receptors as potential binding partners, it is important to profile the expression of different receptor variants in CRC and to assess the effect of receptor activation on CRC progression. Furthermore, the effect of neurotransmitters and neuropeptides on tumor growth could be concentration dependent. This was shown for serotonin, where high doses exerted a mitogenic effect, but low doses reduced growth by limiting the blood flow towards the tumor [61]. Thus, this argues the importance of studying the concentration-dependent effect of other neurotransmitters in the future. It is also noteworthy that some of these signaling molecules are not solely secreted by (enteric) neurons, but can also be produced by other cells, as well as by gut microbiota [62,63,64,65]. Thus, care has to be taken when concluding about the role of neurons in CRC based on neurotransmitter or neuropeptide studies. An overview of the different tumor types, signaling factors, and their (presumed) effect on tumor growth is given in Table 1.

While these modes of communication mostly involve paracrine signaling, exciting new findings, for instance in glioma, show that specific synaptic connections between neurons and cancer cells can be established as well [66,67]. Here, the neurons surrounding the tumor secrete neuroligin-3, which promotes the formation of neuron–glioma synapses with a specific type of glutamate receptors, AMPA-receptors, on the glioma cells [66,67]. Activation of these neurons promotes cancer growth through glutamate release by promoting proliferation and invasion of the glioma cells. Moreover, activation of AMPA-receptors on glioma cells stimulates the release of glutamate from the glioma cells themselves, thereby activating AMPA-receptors in an autocrine fashion and further activating neurons. Although there is currently no evidence for functional synapses between neurons and CRC cells, enteric neurons have been observed in close proximity to tumor cells [40]. Moreover, increased glutamate receptor expression has been associated with CRC [68]. In the future, it would be interesting to study whether synaptic-like connections are formed between neurons and CRC cells.

It is difficult to assess whether the contribution of neurons to colorectal carcinogenesis is attributable to the extrinsic or intrinsic GI innervation. Even though some studies that investigated the structural changes of the ENS in CRC have validated previous findings that CRC is able to induce atrophy of the ENS [69,70], another study has demonstrated that neuronal innervation in the tumor area increases in higher grade tumors [71]. Due to this observation and the known prognostic value of PNI, different researchers studied the prognostic value of other neuronal (related) markers in CRC. As described above, galanin has been suggested as a possible biomarker for CRC in sera and tissue, and has been associated with the recurrence of disease and lower survival rates in stage II CRC patients [72]. By performing immunohistochemistry for tyrosine hydroxylase (TH) and vesicular acetylcholine transferase (VAchT) on CRC tissue, Zhou et al. mainly detected sympathetic (TH^+^) fibers in early phases of CRC, indicating good prognosis, while parasympathetic (VAchT^+^) innervation was observed in later stages and indicated poor prognoses [73]. It should be noted that a vast number of enteric neurons is cholinergic, and thus expresses VAchT [12], raising the possibility that the observed nerve fibers are also of intrinsic origin. Acetylcholine receptors are expressed in CRC tissue in a subpopulation of patients and their expression is associated with a more advanced tumor stage, suggesting that cholinergic parasympathetic nerves could signal to CRC tissue and aggravate disease. Of note, acetylcholine is not necessarily produced only by neurons, but can also be released by colon cancer cells in an autocrine fashion [74], indicating that caution has to be taken when drawing conclusions regarding such neuron–cancer crosstalk observations. In contrast to neural input from the ENS or the extrinsic innervation in CRC, the possibility that cancer cells trans-differentiate into neural(-like) cells has been suggested as well [75]. Cancer stem cells isolated from CRC patients are able to generate neurons in vitro, including sympathetic (TH^+^) neurons positive for synaptic markers (SV2A and synapsin). Moreover, in prostate cancer, it has been shown that doublecortin (DCX^+^) neural progenitors escape the blood–brain barrier to infiltrate the tumor and generate new adrenergic neurons [76]. However, this process has not yet been studied in CRC. It remains to be elucidated how intrinsic, extrinsic, and cancer cell-derived neural input balances in colorectal carcinogenesis. Although the understanding of the role of nerves in cancer is increasing and there are strong indications that nerves are indeed important in CRC, further research is necessary to gain mechanistic insights in the crosstalk between nerves and cancer cells in the context of CRC. 

### 3.2. Glia and Colorectal Cancer

The knowledge about the direct effects of EGCs on colorectal carcinogenesis is still very limited. To study the relationship between CRC cells and EGCs, Valès et al. performed in vitro experiments with those cells [77]. Co-culturing CRC stem cells (CSCs) with EGCs (JUG–EGC cell line) increased the number and size of tumor spheres. The EGCs were cultured on transwell filters and could not directly contact CRC cells, making it likely that EGCs influenced the CSCs via paracrine signaling. Interestingly, the addition of media of EGCs to CSC spheres was only able to increase tumor sphere numbers and size when the EGCs were pre-incubated with CRC cell medium, indicating the importance of bi-directional signaling. Furthermore, the study proposed a model where EGCs activated by CRC medium increased PGE2 release which targeted CSCs and promoted tumor sphere formation via activation of the EP4, EGFR and ERK1/2 pathways. In vivo experiments, injecting human CRC cells with or without EGCs (JUG–EGC cell line) in mice, resulted in increased tumor size in the presence of EGCs. 

An in vivo study has confirmed the tumor-promoting role for EGCs in CRC development [78]. GFAP+ EGCs were depleted in mouse models of CRC (AOM/DSS and Apc^Min/+^) by injecting either ganciclovir in gfap-tk mice or diphtheria toxin in gfap-CRE iDTR mice. Both EGC-depleted mice models showed reduced tumor burden, in number and size. However, once the cancerous lesions had formed, EGC depletion had no effect on tumor burden, indicating that GFAP+ EGCs affect tumor development but not tumor progression. 

Infiltration of EGCs in human CRC tumors was shown by staining for the glial markers S100β and glial fibrillary acid protein (GFAP) [77]. Moreover, Seguella et al. investigated the levels of S100β in colon biopsies of ten CRC patients and eight healthy subjects and showed an increase in S100β expression compared to control, and subsequent activation of the proliferation pathway RAGE/MAPK/NF-κB [79]. Moreover, upregulation of VEGF, IL-6 and AQP4, markers for angiogenesis and invasion, correlated with S100β expression. Based on GFAP immunohistochemistry, EGC density was observed to be the highest in well-differentiated colorectal tumors, and decreased in moderately to poorly differentiated colorectal tumors [80,81]. Furthermore, a high density of GFAP-positive EGCs correlated with reduced proliferation of cancer cells (Ki-67 staining) and attenuated inflammation (CD45 staining) in the tumor tissue [80]. Due to the contradictory findings of these studies, future research will be necessary to explore the exact correlation between EGC density and tumor development/progression. Furthermore, the molecules released by glia and the substances/processes that activate and attract them need to be explored.

### 3.3. Other Members of the Tumor Microenvironment

It is important to realize that, even though neuronal cells and glial cells can influence colorectal carcinogenesis directly, an interplay between these and other cells of the TME will finally determine the fate of a tumor. Immune cells are the most studied members of the tumor microenvironment in multiple cancers, and knowledge about their interaction with extrinsic and intrinsic neurons and glia is rapidly developing [82,83,84,85,86]. In the gut, muscularis macrophages have been shown to interact directly with enteric neurons and EGCs [87,88,89]. How this interaction develops in the context of CRC is not yet known, but an indirect effect of neurons and EGCs on CRC through immune cell regulation is one option. Another link between the ENS and CRC could be the microbiome, which is known to be involved in CRC [90] and interacts with both enteric neurons and EGCs [91,92]. Further studies are needed to unravel the crosstalk between neurons, glia, and other cell types to understand the role of the nervous system in CRC development and progression. In addition, but not specific to CRC, Cole et al. described that the sympathetic nervous system can regulate various processes involved in tumorigenesis, such as proliferation, angiogenesis, and epithelial-mesenchymal transition [93]. In addition, Kuol et al. summarized that various neurotransmitters released from nerve terminals, including serotonin and glutamate, stimulate endothelial cells and the consequent angiogenic process, thereby promoting tumor growth [94]. Lymphangiogenesis can be stimulated in a similar manner, thereby conferring an aggressive advantage to the tumor [95]. Additionally, noradrenaline signaling can, through the activation of β-adrenergic receptors, activate cancer-associated fibroblasts (CAFs) which are able to remodel the extracellular matrix. This remodeling will further support tumor growth [9,96,97,98] or lead to the collapse of blood vessels with a concomitant reduced penetration potential of therapeutics [99]. Thus, these data highlight the importance of the different cell types of the TME for tumorigenesis and suggest that their crosstalk with enteric neurons and EGCs can potentially affect the pathogenesis of CRC.

## 4. Therapeutic Potential

As described above, the presence of nerves and neuronal activity aggravate carcinogenesis in most studies. Consequently, denervation of tumors has been postulated as a therapeutic option for several cancers, including gastric [3] prostate [4] and breast cancer [6]. For CRC, denervation of the myenteric plexus using benzalkonium chloride (BAC) reduced the number of aberrant crypt foci and other indicators of premalignant lesions, suggesting that denervation of the colon in the early stages of CRC could attenuate carcinogenesis [100]. Therapies which target PNI are described in different cancer types, which mostly target NGF–TRKA signaling, because NGF is commonly overexpressed by cancer cells [101]. Small molecule inhibitors of TRKA and antibodies targeting NGF have been used in animal experiments and clinical trials to relieve cancer-associated pain, reverse PNI, and inhibit cancer growth. It remains to be determined if similar strategies would be beneficial for the treatment of CRC as well. While PNI is targeted with radiation therapy in head and neck cancer [102], PNI is not targeted specifically with radiotherapy in CRC. Nonetheless, CRC tumors treated with neoadjuvant chemoradiation also had a decreased occurrence of PNI [38], which seems to contribute to improved prognosis. 

β-adrenergic receptors are also often described as potential therapeutic target for different cancer types, and they are often present on cancer cells [103]. When activated by sympathetic signaling, they can promote metastasis in several cancer types [103]. Whereas β-adrenergic receptors are associated with tumor grade in CRC [71] and meta-analyses of β-blocker use showed improved survival rates in patients with various types of cancer (ovarian cancer, pancreatic cancer, and melanoma), no improved survival time was found for CRC patients [104,105]. Further understanding about the role of the intrinsic and extrinsic GI innervation in carcinogenesis is necessary before new therapeutic options can be revealed, because the research on the role of the nervous system in CRC is still in its infancy. 

## 5. Conclusions and Future Directions

Nerves are increasingly recognized as an important member of the TME, participating in carcinogenesis either (I) passively, serving as guiding strands for tumor spreading; or (II) actively, by secreting trophic factors, neurotransmitters, and neuropeptides that can influence cancer growth. This communication is bi-directional; cancer cells have been shown to induce neuronal innervation and activation of the nervous system can worsen disease outcomes. However, studies regarding the role of the nervous system in CRC remain limited, even though the GI tract is highly innervated by both extrinsic and intrinsic neural networks. Key questions that need to be addressed in future studies include: What are the molecular mechanisms involved in the communication between neural cells and CRC cells? Does neuron/glia–cancer crosstalk occur directly and/or through interplay with other TME components? Which therapeutic options can be considered to intervene in this crosstalk? 

In several animal models of cancer, therapeutic denervation can lead to reduced cancer growth, but it remains to be established whether this is also the case for human subjects with CRC, and to what extent therapeutic denervation of the gut will be possible while retaining all important bowel functions. Identifying to what extent the neural contribution to colorectal carcinogenesis is of extrinsic and/or intrinsic origin imposes another challenge, because these neural networks are extensively connected in the GI tract. Patient stratification based on neural parameters might be valuable in the future for the prognosis or prediction of treatment responses. All in all, more research is necessary to elucidate the importance of the (enteric) nervous system in the pathogenesis of CRC, and new insights could contribute to novel and more targeted treatment options.

## Figures and Tables

**Figure 1 cancers-13-00152-f001:**
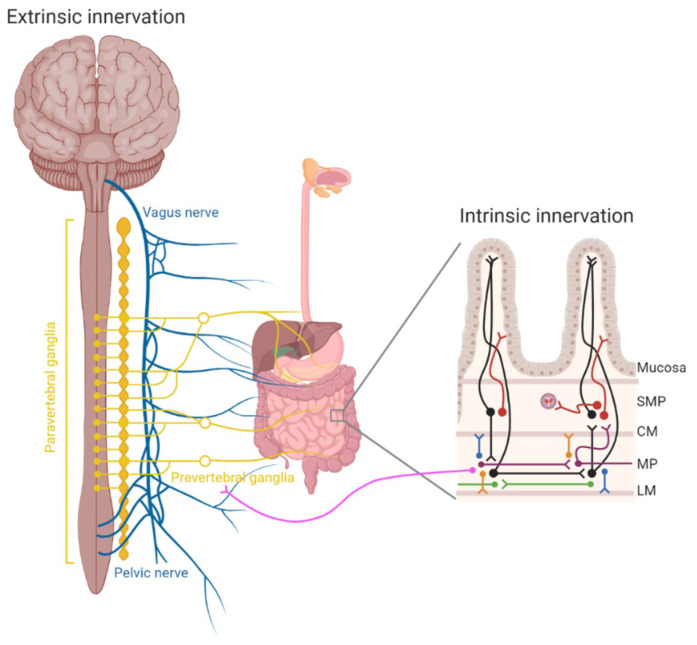
Schematic of the extrinsic and intrinsic innervation of the gastrointestinal tract. Extrinsic innervation comprises both the sympathetic (yellow) and the parasympathetic (blue) nervous system. The sympathetic nervous system has cell bodies located in paravertebral ganglia (next to the spinal cord) and prevertebral ganglia (close to the gastrointestinal tract). The parasympathetic nervous system projects to the stomach, small intestine, colon, and rectum via the vagus nerve and the pelvic nerve. Intrinsic gastrointestinal innervation is provided by the enteric nervous system, which comprises multiple neuronal types clustered within the myenteric plexus (MP) and submucosal plexus (SMP). The major neuronal types in the enteric nervous system are intrinsic primary afferent neurons (sensory neurons with Dogiel type II morphology that project to the mucosa, black), descending (purple) and ascending (green) interneurons, secretomotor and vasodilatator neurons (red), excitatory (blue) and inhibitory (orange) motorneurons, and intestinofugal neurons (pink). SMP, submucosal plexus; CM, circular muscle; MP, myenteric plexus; LM, longitudinal muscle. (Created with Biorender.com).

**Table 1 cancers-13-00152-t001:** The effect of the presence and activity of various nerves and neurotransmitters/peptides on different types of tumor growth.

Type of Innervation	Tumor Type	Neurotransmitters/ Peptides/Activity	Inhibit (−)/Promote (+) Tumor Growth
Extrinsic innervation	Breast cancer [42]	↑ Sympathetic activation	+
	Gastric cancer [3]	↓ Parasympathetic activation	−
	Gastric cancer [49]	↑ Acetylcholine transferase ACh → Nerve growth factor (NGF)	+
	Intestinal cancer [43]	(Parasympathetic) vagotomy Sympathetic denervation	− /
	Pancreas cancer [48]	(Sympathetic) adrenergic signaling → NGF → Brain-derived neurotrophic factor (BDNF)	+
	Glioma [66,67]	Neuroligin-3 → glutamate	+
	Colorectal cancer (CRC) [69]	Sympathetic fibers in early tumors	−
	CRC [69]	(Parasympathetic?) cholinergic fibers in late-stage tumors	+
Unknown—intrinsic innervation?	CRC [50]	Substance P	+
	CRC [51,52]	Neurotensin	+
	CRC [53]	Somatostatin	−
	CRC [54,55]	Galanin	biomarker
	CRC [58]	Neuropeptide Y (NPY)	+
	CRC [59]	Serotonin	−/+
	CRC [60]	gamma-aminobutyric acid (GABA)	−(/+)

## Data Availability

No new data were created or analyzed in this study. Data sharing is not applicable to this article.

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
