# Peer review of "The Emerging Role of Nerves and Glia in Colorectal Cancer"

_cancers, 2021, doi:10.3390/cancers13010152_

Round 1

Reviewer 1 Report

This review summarized the organization of the extrinsic and intrinsic nervous system innervation in the GI tract, underscored the reciprocal roles of neurons/EGCs and CRC played on each other, and highlighted the therapeutic potentials of manipulating nervous system innervation in CRC. This work is well organized, and will make a great overview of the field. I have the following comments to help improve it:  

  1. Though introducing some background knowledge of the innervation of the gastrointestinal tract is necessary, I feel that section 2 is too detailed and a bit long. I would suggest deleting some irrelevant part. For example, there is a whole paragraph talking about EGC subtypes, which I think is not very important in this context.
  2. Would it be possible to add some schematics to summarize section 3? It would be extremely helpful.
  3. It would be great to add a section that summarizes key questions for future research. Though this has been discussed a little bit along the way, it would be nice to have a separate part dedicated for future outlook.

Minor point:

Line 252:

Although enteric neurons have been observed in close proximity to tumor cells [51], there is currently no evidence for functional synapses between neurons and CRC cells. Moreover, increased glutamate receptor expression has been associated with CRC

I suggest changing the order of the sentences to make it more logical, maybe something like: Although there is currently no evidence for functional synapses between neurons and CRC cells, enteric neurons have been observed in close proximity to tumor cells. Moreover, increased glutamate receptor expression has been associated with CRC. In the future, it would be interesting to look if synaptic connections form between neurons and CRC cells.

Author Response

We would like to thank reviewer 1 for taking the time to thoroughly assess our literature review and are happy to see that our work is appreciated.

  1. We agree that section 2 contains many details and therefore we have shortened this section in the following way:
  • removed: “in the myenteric plexus” and “via the pelvic plexus” (line 90-91)
  • removed: glial cell types in extrinsic innervation part (line 97-100)
  • removed: details about neuronal subtypes in intrinsic innervation part (line 131-144)
  • removed as suggested: glial subtypes in intrinsic innervation part (line 147-159)
  1. To provide a comprehensive overview of section 3, we have added a table summarizing the information described in this section. The table (table 1) is referred to in line 244-245 and contains an overview of the different tumor types, the presence and activity of various nerves and neurotransmitters/peptides, and the effect on tumor growth and aggressiveness.
  2. We have elaborated on and added key questions that should be addressed in future research in the final section and altered the heading to ‘Conclusions and future directions’.

Added:

“Key questions that need to be addressed in future studies include:  What are the molecular mechanisms involved in the communication between neural cells and CRC cells? Does neuron/glia-cancer crosstalk occur directly and/or through the interplay with other TME components? Which therapeutic options can be considered to intervene in this crosstalk?

In several animal models of cancer, denervation can lead to reduced cancer growth, but it remains to be established whether this is also the case for human subjects with CRC, and to what extent therapeutic denervation of the gut will be possible while retaining all important bowel functions.” (line 379-386)

“Patient stratification based on neural parameters might be valuable in the future for prognosis or prediction of treatment response.” (line 388-389)

Minor point:

We altered the sentence (now line 254) accordingly.

Reviewer 2 Report

Dear Authors, the research field of your review is interesting and opens new aspects in the field of interactions between cancer development and microenvironment. Researchers have been very enthusiastic about the CRC and its relationship, just look at the role of the intestinal microbiota. Unfortunately I believe that the data in the literature that can be reported in a review with a title like yours are still few and not significant.

Author Response

We thank reviewer 2 for assessing our manuscript. We agree that data on the role of the nervous system in CRC are still limited and that the field is still in its infancy, which is one of the reasons why we incorporated studies on other cancer types as well, in an attempt to mirror them with the pathophysiology of CRC. However, in agreement with the evaluation of the other reviewers, we believe our review of the literature is timely as the field is rapidly expanding and gaining attention. To concur with reviewer’s remark, we have amended the title of our manuscript to “The emerging role of nerves and glia in colorectal cancer”, which should better reflect the novelty of this topic. In addition, we have incorporated new studies that have been recently published on the role of enteric glial cells in the revised version of this manuscript (line 304-317). Finally, we expanded the conclusion section to further emphasize that many questions on the role of the nervous system in CRC remain to be answered in future research.

Reviewer 3 Report

This is a well written and timely review that will be useful tool for CRC researchers and neuroscientists alike. I have just 3  recommendations: 

1) The diagram of the extrinsic and intrinsic innervation of the gut is useful, but because most CRC researchers are not familiar with different neural types I think a table would useful. This could break down into different categories and link neurons to potential signaling molecules.

2) Another table listing which signals have been linked to which cancers, and whether they are tumor promoting or tumor inhibiting would also be useful.

3) The paragraph on therapeutic potential is somewhat sparse. Is there a potential to inhibit metastasis by targeting mechanisms in PNI?

Author Response

We are thankful for the suggestions from reviewer 3 to improve our manuscript. We have made the following changes:

  1. We understand that all the neuronal subtypes and neurotransmitters mentioned can become confusing to cancer researchers. As suggested by reviewer 1, we therefore removed some parts in section 2 that contained too many details. In the same line, we believe it is not within the scope of the current review to give an extensive overview of all the current neuronal subtypes, their respective neurochemical coding and detailed neurotransmitter repertoire. This level of detail can be found in dedicated review articles to which we refer in line 131 [26,30,31]. Nonetheless, we have added a table to summarize the evidence on the role of specific nerve cells and/or their mediators in CRC (please also see point 2).
  2. As suggested by the reviewer, we have included a table (table 1) containing a summary of section 3 (line 259).
  3. We have expanded the paragraph on therapeutic potential by adding:
    Therapies which target PNI are described in different cancer types, which mostly target NGF-TRKA signaling, as NGF is commonly overexpressed by cancer cells [110]. Small molecule inhibitors of TRKA and antibodies targeting NGF have been used in animal experiments and clinical trials to relieve cancer-associated pain, reverse PNI, and inhibit cancer growth. It remains to be determined if similar strategies would be beneficial for the treatment of CRC as well. While PNI is targeted with radiation therapy in head and neck cancer [111], PNI is not targeted specifically with radiotherapy in CRC. Nonetheless, CRC tumors treated with neoadjuvant chemoradiation also had a decreased occurrence of PNI [51], which seems to contribute to improved prognosis.” (line 354-362).
    Unfortunately, data about the targeting of nerves/PNI in CRC is very limited, which is why we incorporated targeting of nerves/PNI in other cancer types and tried to link that with CRC.

Reviewer 4 Report

In this review, Schonkeren and co-authors summarize the role of enteric nervous system in regulating the growth of colon cancer and the potential to target it for therapy. Specifically, the authors focus on the role of glial cells in influencing the tumor microenvironment in the gut, with the role of ENS covered in their previously published review in 2017.

The authors should discuss the following points to improve the manucript:

  1. Is the correlation between PNI and aggressiveness of CRC causative, or simply a result of retrospective studies. It would be beneficial to cite that data is available.
  2. What level of neurotransmitter secretion from the ENS to cancer tissue leads to an effect on its growth and aggressiveness? Is this information available, especially as a quantitative measurement?
  3. Can the authors speculate on the reasons for high levels of EGC leading to lower tumor growth? This process would presumably be different from the role of glial cells in the central nervous system, so this is an interesting facet of glial biology which should be explored in more detail.
  4. How does the therapeutic denervation of the gut lead to reduced CRC growth?

Author Response

We appreciate the review of our manuscript by reviewer 4 and have attempted to answer all of the questions.

  1. So far, the correlation between PNI and aggressiveness of CRC has mostly been investigated in retrospective studies. Indeed, a correlation should not be confused with causation. Functional experiments will have to be performed to prove a causative relation.
  2. As suggested by the reviewer, neurotransmitters can have a concentration-dependent effect on tumor growth and aggressiveness. We have described one such effect for serotonin (line 238-241). However, most studies unfortunately do not report concentration ranges to study effects on tumor growth or aggressiveness. In addition, while in vitro studies might be instrumental to establish concentration-dependent effects, we believe it is very challenging to detect, let alone quantify, local levels of neurotransmitters in vivo, as their concentrations usually rapidly fluctuate depending on activity, half-life and micro-environment (e.g. at presumptive tumor-nerve ‘synapses’).
  3. Currently this is not yet clear, as studies are contradictory. However, we added two new studies to the review (line 304-315) and most studies actually suggest that high levels of EGCs lead to more tumor growth.
  4. We agree that the hypothesis that therapeutic denervation will lead to reduced colorectal tumor growth is interesting. However, to the best of our knowledge the only study directly investigating this (Vespúcio et al. 2008), describes denervation of the myenteric plexus in a mouse model of CRC and reports a reduced number of premalignant lesions. The exact mechanisms involving the reduction in tumorigenesis are not yet clear. Unfortunately, this question also remains to be answered in human patients, as studies investigating this are lacking. To highlight this knowledge gap, we have added the following to the conclusions and future directions section: “In several animal models of cancer, denervation can lead to reduced cancer growth, but it remains to be established whether this is also the case for human subjects with CRC, and to what extent therapeutic denervation of the gut will be possible while retaining all important bowel functions.” (line 383-386). Because of the lack of experimental evidence, we believe it is too early to further speculate about this matter and the mechanisms involved.